# Design and Analysis of Ultra-Thin Broadband Transparent Absorber Based on ITO Film

**DOI:** 10.3390/mi16060653

**Published:** 2025-05-29

**Authors:** Zibin Weng, Yahong Li, Youqian Su, Zechen Li, Jingnan Guo, Ziming Lv, Chen Liang

**Affiliations:** The National Key Laboratory of Radar Detection and Sensing, Xidian University, Xi’an 710071, China; zibinweng@mail.xidian.edu.cn (Z.W.); 22021211699@stu.xidian.edu.cn (Y.S.); 23021211560@stu.xidian.edu.cn (Z.L.); 18066795882@163.com (J.G.); 23021211267@stu.xidian.edu.cn (Z.L.); liangchen@stu.xidian.edu.cn (C.L.)

**Keywords:** ITO film, transparent absorber, interference theory, ECM

## Abstract

In this paper, we design an ultra-thin broadband transparent absorber based on indium tin oxide (ITO) film, and we choose polymethyl methacrylate (PMMA) high-transmittance dielectric sheet instead of the traditional dielectric sheet and polyethylene glycol terephthalate (PET) as the ITO film substrate. Simulation results indicate that the absorber achieves more than 90% absorption for positively incident electromagnetic waves in the broadband range of 5–21.15 GHz with a fractional bandwidth (FBW) of 123.5% and a thickness of 6.3 mm (0.105 *λ_L_*, where *λ_L_* is the wavelength at the lowest frequency). Meanwhile, this paper introduces the interference theory to explain the broadband absorption mechanism of the absorber, which makes up for the defect that the equivalent circuit model (ECM) method cannot analyze the oblique incidence electromagnetic wave. This paper also compares the HFSS simulation results, ECM theoretical values, and interference theoretical values under positively incident electromagnetic waves to clarify the advantages of interference theory in the design of wave absorbers.

## 1. Introduction

With the rapid development of 5G technology and wireless communications, the increasingly complex electromagnetic environment has put forward higher requirements for wave-absorbing materials. As a key material for solving electromagnetic interference problems, the core application scenarios of wave absorbers include improving the electromagnetic compatibility (EMC) of electronic systems, reducing the radar cross-section (RCS), and suppressing electromagnetic pollution [1,2,3]. We can divide wave absorbers into two main types based on the differences in their material systems and mechanisms of action: One type uses traditional materials as wave absorbers, such as graphite, carbon black, ferrite, etc., through the coating or impregnation process to form a loss layer. This type relies on dielectric or magnetic loss mechanisms to convert incident electromagnetic waves into heat or other forms of energy, thereby suppressing surface electromagnetic wave reflection [4,5,6,7]. The other category is metamaterial absorber (MMA), which is based on the design of artificial periodic structure and realizes the efficient absorption in wide bandwidth through the synergy of multiple mechanisms, such as impedance matching, resonance loss, and interference phase cancellation. Compared with traditional wave-absorbing materials, MMA breaks through the “*λ*/4” thickness limitation and achieves strong absorption at sub-wavelength thickness through localized resonance. In addition, MMA’s ability to flexibly adjust unit structure parameters helps to optimize electromagnetic impedance matching and resonance loss characteristics. This provides a new degree of design freedom for the development of high-performance absorbers [8,9,10].

As early as 2008, Landy, N.I. et al. proposed an MMA based on metal units and FR4 dielectric plates, which achieved near-perfect absorption at an operating frequency of 11.48 GHz, and since then, MMA has attracted a lot of attention [11]. In 2009, Zadeh, A.K. et al. proposed a fast method for designing wave absorbers based on ECM [12]. In 2019, Xie, J et al. designed an all-dielectric MMA to achieve 0–45° angular domain absorption for two polarization waves [13]. In addition, the rapid development of hypersurface technology provides new ideas for MMA design [14,15]. Zheng, C.-L. et al. designed a multifunctional hypersurface for polarization multiplexing by jointly regulating the propagation phase and the geometric phase. This study demonstrated the flexible ability of the hypersurface to regulate the phase, polarization, and energy of electromagnetic waves, and its design concept is also applicable to the optimization of MMA [16]. The above designs have contributed to the development of MMA, but the use of opaque metals and dielectrics has made these designs difficult to apply where high optical transparency is required.

In recent years, with the rapid development of transparent materials, many people use transparent materials in combination with wave-absorbing materials. They have made MMA optically transparent while combining the characteristics of “thin thickness, light weight, wide bandwidth, and strong wave-absorbing ability” [17,18,19,20,21]. Xiong, Y. et al. designed a transparent absorber in the form of ITO-PMMA-ITO by utilizing ITO film instead of metal and PMMA as the dielectric plate, which can achieve more than 90% wave absorption in the frequency range of 6–17.8 GHz [22]. This transparent MMA designed based on ITO film can still be analyzed by ECM. Dong, L. et al. designed an optically transparent MMA based on ECM, which can absorb more than 90% of electromagnetic waves in the frequency range of 15.77–38.69 GHz [23]. Lai, S. et al. have also designed a transparent absorber in the form of ITO-PET-ITO based on ITO film and PET dielectric using an equivalent circuit model, which absorbs more than 90% of the wave in the frequency range of 19.9–44.6 GHz [24]. While ECM remains prevalent in MMA design, its foundational assumptions fail to account for oblique incidence scenarios. Therefore, an analytical method based on the theory of interference has been proposed, which allows obtaining reflection coefficients related to the electromagnetic parameters and the thickness of the medium [25,26]. Starting from the multiple-reflection interference model, Chen, H.-T. demonstrated that the near-field interaction or magnetic response between neighboring metal structures in MMA is negligible, and the two layers can be decoupled, with the only connection being multiple reflections between them [27]. Zeng, X. et al. designed a three-band MMA based on ECM and interference theory, which exhibits high absorption greater than 95% at 4.4 GHz, 6.05 GHz, and 13.9 GHz. Theoretical predictions and simulations, as well as experimental measurements, are in excellent agreement with each other, which also proves the feasibility of the interference theory [28].

In MMA design, the electromagnetic parameters and thickness of the medium significantly affect the propagation phase of electromagnetic waves. When the phase difference between two electromagnetic waves is an odd multiple of π, the interference phase cancellation phenomenon occurs, thus realizing efficient wave absorption. In this paper, the interference theory of monolayer dielectric is extended to the multilayer dielectric system, and a transparent absorber is designed using ITO film with PMMA and PET transparent dielectric sheets. By comparing the multilayer interference theory predictions with the HFSS full-wave simulation results, we find that there is a non-negligible weak electromagnetic coupling effect between the ITO film layers. This study also compares and analyzes the reflection coefficients obtained by three different methods (multilayer interference theory, ECM, and HFSS full-wave simulation) under positive incidence conditions and clarifies the advantages of interference theory in the design of wave absorbers. The schematic diagram of the transparent absorber in this paper is shown in Figure 1, which has an absorption rate of more than 90% for positively incident electromagnetic waves in a wide frequency range of 5–21.15 GHz. In the case of oblique incidence, it maintains a high absorption rate of more than 90% for TM-polarized waves in the range of 5.35–21 GHz and TE-polarized waves in the range of 5.8–20.65 GHz at the angle of incidence of 0–45°. The design of this absorber, which combines broadband, wide-angle domain, and optically transparent properties, provides an important solution for next-generation stealth technology and electromagnetic compatibility applications.

## 2. Design and Analysis

### 2.1. Modular Design

Figure 2a demonstrates the four-layer symmetric-structure absorber unit designed in this study, where the symmetric nature of the unit pattern gives the structure the advantage of polarization insensitivity. In order to realize the synergistic design of broadband wave absorption and optical transparency, PMMA with dielectric constant *ε*_1_ = 3.05 − j0.0305 is selected as the transparent dielectric plate for the first and third layers. The second-layer structure is shown in Figure 2b, where an ITO thin film pattern with a surface resistance *I*_1_ = 80 Ω/sq is prepared by an etching process on a highly transparent PET substrate with a dielectric constant *ε*_2_ = 3 − j0.18 to form a lossy layer. The fourth layer adopts a highly conductive ITO film (*I*_2_ = 6 Ω/sq) to replace the traditional metal reflector plate, the top view of which is shown in Figure 2c, to ensure the electromagnetic wave reflecting performance while maintaining the optical transparency characteristics.

Figure 3a illustrates the evolution of the MMA cell, and in order to compare the reflection of electromagnetic waves by each structure, we make all the structural parameters consistent with Figure 2. Circular resonators are known to be widely used in MMA designs and exhibit excellent polarization insensitivity due to the geometrical symmetry of their structures. However, the analysis of the reflection coefficient curve in Figure 3b shows that Structure A can only cover the 7.5–20 GHz band at −10 dB operating bandwidth, with two strong resonance points at 12.2 GHz and 14.6 GHz, respectively. In order to expand the wave-absorbing bandwidth, it is necessary to realize the redistribution of resonance modes through structural optimization and to shift the double resonance point to the direction of low frequency and high frequency in a synergistic way.

After we reconfigured the initial cell into a circular resonator (Structure B), its dual resonance points were shifted to 8.1 GHz and 19.7 GHz. However, as shown in Figure 3b, the structure exhibits a significant absorption depression in the 9.5–18.8 GHz band, which is attributed to the lack of ohmic loss due to the missing ITO film in the center region of the ring and the limited electromagnetic coupling efficiency of the single ring resonator. To solve the above problem, we introduce circular patches of radius *r*_3_ in the structure C. Experimental data show that the −10 dB bandwidth of Structure C extends to 5.6–19.9 GHz, but the depth of absorption in the lower frequency band is insufficient. We finally succeeded in exciting the dual resonant modes at 6.5 GHz and 17.9 GHz by implementing a slotting design (Structure D) in the outer ring arm, and the optimized Structure D achieves highly efficient absorption (≥90% absorption) in the 5–21.15 GHz band.

In order to deeply investigate the modulation mechanism of MMA unit structure parameters on the electromagnetic wave reflection characteristics, we systematically analyzed the reflection coefficients under different parameter conditions based on ANSYS HFSS 2020R2. Figure 4a,b demonstrate the pattern of the radius parameters *r*_2_ and *r*_3_ on the reflection coefficient, respectively. As *r*_2_ gradually decreases from 6 mm to 5 mm, the width of the central annular slit decreases accordingly, leading to a significant increase in the depth of reflection at both resonance points. This phenomenon suggests that a moderate narrowing of the gap can effectively enhance the local coupling effect of the electromagnetic field. It is worth noting that an increase in *r*_3_ is physically equivalent to a decrease in *r*_2_, and thus the two are consistent in their mechanisms of modulation of MMA performance. When *r*_2_ = 4.5 mm, the intermediate gap is completely closed, at which time the structural properties are similar to those of structure A in Figure 3a, with the low-frequency resonance point gradually shifted to the high frequency and the reflection depth reduced. After comprehensive optimization, we finally selected *r*_2_ = 5 mm and *r*_3_ = 4.5 mm as the best parameter combination.

The effect of the spacing parameter *s* on the reflection coefficient is further investigated in Figure 4c. As *s* decreases from 6 mm to 2 mm, the circular gap gradually becomes narrower, and the wave-absorbing property is subsequently enhanced. However, when *s* = 0 mm (the circular gap is completely closed), the depth of reflection at the low-frequency resonance point decreases significantly, which destroys the original resonance characteristics. After weighing the bandwidth against the absorption depth, we determined that *s* = 2 mm is the optimal value. The above results indicate that the variation of the structural parameters of the MMA unit can significantly modulate its resonance characteristics, which is one of the key mechanisms to realize the broadband efficient absorption.

The broadband absorption mechanism of this MMA is not only dependent on the optimization of structural parameters but also closely related to the ohmic loss of the ITO film. Among them, the surface resistance value (*I*_1_) of the first ITO film has a significant modulation effect on the reflection characteristics, as shown in Figure 4d. With the increase of *I*_1_ from 40 Ω/sq to 100 Ω/sq, the low-frequency resonance point is shifted to the high-frequency direction, and the depth of the reflection coefficient is obviously increased, and the resonance effect in the high-frequency band is also enhanced. However, this variation simultaneously leads to a decrease in the absorption performance in the center band. Through comparative analysis, it is found that the −10 dB relative bandwidth of the MMA reaches the optimal value of 123.5% when *I*_1_ = 80 Ω/sq, which achieves a balance between wide bandwidth and strong absorption. 

### 2.2. Interference Theory Analysis

Interference phase cancellation is another key mechanism to realize broadband wave absorption in this design. When the phase difference between two columns of electromagnetic waves is an odd multiple of π, the interference phase cancellation phenomenon will occur. The total reflection coefficient of a monolayer medium obtained by the interference theory is shown in Equation (1) [25]:(1)$$\begin{array}{c} S_{11\_total}=S_{11}+S_{21}e^{-j\beta }e^{-j\pi }e^{-j\beta }S_{12}\\ \;+S_{21}e^{-j\beta }e^{-j\pi }e^{-j\beta }{\left(S_{22}e^{-j\beta }e^{-j\pi }e^{-j\beta }\right)}^{1}S_{12}\\ \;+S_{21}e^{-j\beta }e^{-j\pi }e^{-j\beta }{\left(S_{22}e^{-j\beta }e^{-j\pi }e^{-j\beta }\right)}^{2}S_{12}+\ldots \\ \;=S_{11}+S_{21}e^{-j(2\beta +\pi )}S_{12}\sum_{n=0}^{\infty }{\left(S_{22}e^{-j(2\beta +\pi )}\right)}^{n}\\ \;=S_{11}+\frac{S_{21}S_{12}e^{-j(2\beta +\pi )}}{1-S_{22}e^{-j(2\beta +\pi )}} \end{array}$$
where $S_{(n-1)(n-1)}=(\eta_{n}-\eta_{n-1})/(\eta_{n}+\eta_{n-1})$, $S_{n(n-1)}={2\eta }_{n}/(\eta_{n}+\eta_{n-1})$, $\eta_{n}=\sqrt{\mu_{n}/\varepsilon_{n}}$, $\beta =(2\pi \mathrm{fd}/c)\sqrt{\mu_{i}\varepsilon_{i}}$, *c* is the speed of light, *f* is the frequency, and *d* is the thickness of the medium. Table 1 shows the expressions of reflection coefficient and transmission coefficients.

In this study, due to the design of multilayer dielectric structures, the above interference theory needs to be extended to multilayer systems. Figure 5 illustrates the propagation of electromagnetic waves in a multilayer medium, whose total reflection coefficient $S_{11\text{\_}\mathrm{total}}$ can be calculated by Equation (2):(2)$$\begin{array}{c} S_{11\_total}=S_{11}+\frac{S_{21}S_{12}K_{1}e^{-j2\beta_{1}}}{1-S_{22}K_{1}e^{-j2\beta_{1}}}\\ K_{1}=S_{22}+\frac{S_{32}S_{23}K2e^{-j2\beta_{2}}}{1-S_{33}K2e^{-j2\beta_{2}}}\\ K_{2}=S_{33}+\frac{S_{43}S_{34}K3e^{-j2\beta_{3}}}{1-S_{44}K3e^{-j2\beta_{3}}}\\ \ldots \ldots \\ K_{n-1}=S_{\mathrm{nn}}+\frac{S_{(n+1)n}S_{n(n+1)}e^{-j(2\beta_{n+1}+\pi )}}{1-S_{\left(n+1)(n+1\right)}e^{-j(2\beta_{n+1}+\pi )}} \end{array}$$

When an electromagnetic wave is irradiated to the surface of a medium by oblique incidence, it can be decomposed into two orthogonal polarization components: perpendicularly polarized wave (TE) and parallel polarized wave (TM), where the reflection coefficient $S_{11⊥\;}$ and transmission coefficient $S_{21⊥\;}\;$ of the perpendicularly polarized wave (TE) are, respectively, [29]:(3)$$S_{11⊥}=\frac{\eta_{1}\cos\theta_{i}-\eta_{0}\cos\theta_{t}}{\eta_{1}\cos\theta_{i}+\eta_{0}\cos\theta_{t}}=\frac{\cos\theta_{i}-\sqrt{\frac{\mu_{0}\varepsilon_{1}}{\mu_{1}\varepsilon_{0}}-\frac{\mu_{0}^{2}}{\mu_{1}^{2}}\sin^{2}\theta_{i}}}{\cos\theta_{i}+\sqrt{\frac{\mu_{0}\varepsilon_{1}}{\mu_{1}\varepsilon_{0}}-\frac{\mu_{0}^{2}}{\mu_{1}^{2}}\sin^{2}\theta_{i}}}$$(4)$$S_{21⊥}=\frac{2\eta_{1}\cos\theta_{i}}{\eta_{1}\cos\theta_{i}+\eta_{0}\cos\theta_{t}}=\frac{2\cos\theta_{i}}{\cos\theta_{i}+\sqrt{\frac{\mu_{0}\varepsilon_{1}}{\mu_{1}\varepsilon_{0}}-\frac{\mu_{0}^{2}}{\mu_{1}^{2}}\sin^{2}\theta_{i}}}$$

The reflection coefficient ${\;S}_{11//\;}\;$ and transmission coefficient ${\;S}_{21//\;}\;$ of the parallel polarized wave (TM) are: (5)$$S_{11//}=\frac{\eta_{0}\cos\theta_{i}-\eta_{1}\cos\theta_{t}}{\eta_{0}\cos\theta_{i}+\eta_{1}\cos\theta_{t}}=\frac{\cos\theta_{i}-\sqrt{\frac{\mu_{1}\varepsilon_{0}}{\mu_{0}\varepsilon_{1}}-\frac{\varepsilon_{0}^{2}}{\varepsilon_{1}^{2}}\sin^{2}\theta_{i}}}{\cos\theta_{i}+\sqrt{\frac{\mu_{1}\varepsilon_{0}}{\mu_{0}\varepsilon_{1}}-\frac{\varepsilon_{0}^{2}}{\varepsilon_{1}^{2}}\sin^{2}\theta_{i}}}$$(6)$$S_{21//}=\frac{2\eta_{1}\cos\theta_{i}}{\eta_{0}\cos\theta_{i}+\eta_{1}\cos\theta_{t}}=\frac{2\sqrt{\frac{\mu_{1}\varepsilon_{0}}{\mu_{0}\varepsilon_{1}}}\cos\theta_{i}}{\cos\theta_{i}+\sqrt{\frac{\mu_{1}\varepsilon_{0}}{\mu_{0}\varepsilon_{1}}-\frac{\varepsilon_{0}^{2}}{\varepsilon_{1}^{2}}\sin^{2}\theta_{i}}}$$

When an electromagnetic wave is directed obliquely to the surface of a medium at an angle of incidence $\theta_{i}$, it can be shown that $\theta_{t1}$ is $arcsin(\sqrt{\left(\mu_{0}\varepsilon_{0})/(\mu_{1}\varepsilon_{1}\right)}sin\theta_{i1})$, according to $sin\theta_{t}/sin\theta_{i}=\sqrt{\left(\mu_{0}\varepsilon_{0})/(\mu_{1}\varepsilon_{1}\right)}$. The actual path of the electromagnetic wave in the first layer of the medium is $d_{1}/cos(arcsin(\sqrt{\left(\mu_{0}\varepsilon_{0})/(\mu_{1}\varepsilon_{1}\right)}sin\theta_{i1}))$ and so on to obtain the angle of incidence $\theta_{in}$, the angle of refraction $\theta_{tn}$, and the actual path *d_n_* for each layer.

To reveal the broadband absorption mechanism of the absorber, this study uses a combination of interference theory modeling and full-wave simulation for quantitative analysis. According to the findings of the literature [25], the near-field coupling effect between the ITO film pattern and the reflector plate in the absorber structure is negligible. Based on this theoretical framework, in this study, the analytical solutions of the reflection and transmission coefficients are calculated by considering the three parts of PET (*d*_2_ = 0.125 mm), ITO film (*I*_1_ = 80 Ω/sq), and PMMA (*d*_3_ = 3 mm) as a whole, which are simulated individually by HFSS software, and the analytical solutions of the reflection and transmission coefficients are substituted into the formulae of the multilayer interference theory (Equation (2)).

Figure 6 demonstrates the comparative analysis of the HFSS full-wave simulation results with the multilayer interference theory model. Among them, the yellow solid line indicates the reflection characteristics of the MMA for electromagnetic waves with different incidence angles shown in Figure 2a, while the red solid line is the result of the theoretical calculation based on the multilayer interference theory. It is shown that when a low-surface-resistance ITO reflective layer (square resistance *I*_2_ = 6 Ω/sq) is used, the simulation results show a significant trend of agreement with the theoretical predictions, but the theoretical model predicts significantly deeper reflection coefficient valleys than the simulation results at the resonance frequency point. This discrepancy stems from the fact that the theoretical model approximates the low-resistance ITO layer as an ideal electrical conductor (PEC), while the actual ITO film (6 Ω/sq) does not completely suppress electromagnetic wave transmission.

To verify this mechanism, this study further replaces the underlying ITO with a PET substrate for comparative simulation. The results show that the reflection characteristics of the green solid line (PEC replacement group) and the red dashed line (theoretical value) exhibit a high degree of agreement in the two polarization bands, and this control experiment conclusively demonstrates that the non-ideal reflection due to the finite conductivity is the main reason for the difference. At the same time, this result also reveals the weak electromagnetic coupling effect between the upper and lower ITO film layers. 

In order to further verify the correctness of the interference theory, we analyze it in comparison with the theoretical values of ECM. By analyzing the simulated electromagnetic field distribution on the surface of the ITO film at the resonance points (6.5 GHz and 17.9 GHz), the loss mechanism of the absorber can be deeply analyzed to construct the EMC. As shown in Figure 7, at 6.5 GHz and 17.9 GHz, the electric field energy is highly concentrated at the edge of the inner and outer rings and the slit region, indicating that the equivalent capacitance is formed here due to charge aggregation. The magnetic field is mainly concentrated in the outer annular ring arms, as well as in the center region of the inner circle, whose time-varying properties induce an equivalent inductance. The synergistic effect of the above electric and magnetic fields can be modeled as a second-order R-L-C resonant circuit. Based on the transmission line theory, the PMMA layer and the PET layer can be equated to a transmission line with characteristic impedances of *Z_PMMA_* and *Z_PET_*, and the constructed EMC is shown in Figure 8a.

Based on the unit dimensions in Figure 1, the Advanced Design System 2024 was used to optimize the parameters of the ECM, and the final circuit component values were determined as *R*_1_ = 256.217 Ω, *R*_2_ = 222.254 Ω, *C*_1_ = 5.20518 fF, *C*_2_ = 34.0447 fF, *L*_1_ = 13.0811 nH, and *L*_2_ = 22.6156 nH.

The absorbance of the absorber is shown in Equation (7):(7)$$A=1-T-R=1-|S_{11}{|}^{2}-|S_{21}{|}^{2}$$

Figure 8b compares the HFSS simulation results, the ECM theoretical values, and the multilayer interference theory values of the MMA under positive incidence conditions of electromagnetic waves. The results show that compared to the ECM theoretical values, the interferometric theoretical values match better with the HFSS simulation results in the overall trend, but the weak coupling effect between the ITO films above and below the MMA causes the ECM theoretical values to be closer to the HFSS simulation results in terms of reflection depth.

Based on the multilayer interference theory, it is known that the dielectric constant *ε* and the thickness *d* of the dielectric layer are two key parameters affecting the performance of the absorber. Since the sub-wavelength thickness of the PET layer has less influence on the phase of electromagnetic wave propagation, we mainly analyze the influence of electromagnetic parameters and the thickness of the PMMA layer on the performance of the absorber. Based on the HFSS simulation data in Figure 9, it can be seen that the two resonance points are gradually shifted to lower frequencies as *ε*_1_ gradually increases. Based on the HFSS simulation data in Figure 10, it can be seen that, keeping *d*_3_ constant, the resonance points at high frequencies are gradually formed as *d*_1_ increases, and the two resonance points are gradually shifted to low frequencies. Keeping *d*_1_ constant, as *d*_3_ increases, the resonance point at the high frequency first forms and then disappears, and the resonance point at the low frequency is similarly shifted to the low frequency. When both *d*_1_ and *d*_3_ are 3 mm, both resonance points are formed and the −10 dB bandwidth is more desirable, which proves that at this time, the electromagnetic wave occurs many times in the internal interference phase cancellation of the absorber. Meanwhile, the high similarity between the TM polarization wave and TE polarization wave results verifies that the designed MAA is insensitive to polarization.

## 3. Experimental Verification

To verify the broadband absorption properties of the proposed absorber, we prepared the experimental samples shown in Figure 11a. The sample consists of a 10 × 10 array of periodic cells with an overall size of 150 mm × 150 mm × 6.3 mm. The YHT-23 transmittance tester (produced in China Guangdong Shenzhen YuanHengTong Technology Co., Shenzhen, China) was used to characterize the optical properties of the samples, and the test results are shown in Figure 11b. The average transmittance data of different wavelengths are obtained through multi-point sampling measurements: the transmittance in the UV band (365 nm) is the lowest at 6.5%; the transmittance in the visible band (530 nm) is increased to 51.3%; and the near-infrared band (940 nm) shows the best transmittance performance at 64.7%.

The experimental verification of the designed wave absorber is carried out, and the simulation and experimental results are shown in Figure 12. In the case of positive incidence, the absorber achieves more than 90% absorption in the broadband range of 5–21.15 GHz; in the case of oblique incidence, the absorber achieves more than 90% absorption in the angular domains of 0–45° for TM-polarized waves in the range of 5.35–21 GHz and in the angular domains of 0–45° for TE-polarized waves in the range of 5.8–20.65 GHz. Table 2 shows a comparison of the data from this work with MMA designs from recent years.

## 4. Conclusions

In this paper, an ultra-thin broadband transparent absorber is designed based on ITO film and PMMA and PET transparent media, and the parameters of this absorber are analyzed based on multilayer interference theory. Simulation results indicate that the absorber achieves more than 90% absorption in the broadband range of 5–21.15 GHz for forward-incidence electromagnetic waves; for oblique-incidence electromagnetic waves, the absorber achieves 0–45° angular domain absorption for TM-polarized waves in the range of 5.35–21 GHz and TE-polarized waves in the range of 5.8–20.65 GHz, and the experimental results are in good agreement with the simulation results. This work promotes MMA’s move toward broadband and transparency.

## Figures and Tables

**Figure 1 micromachines-16-00653-f001:**
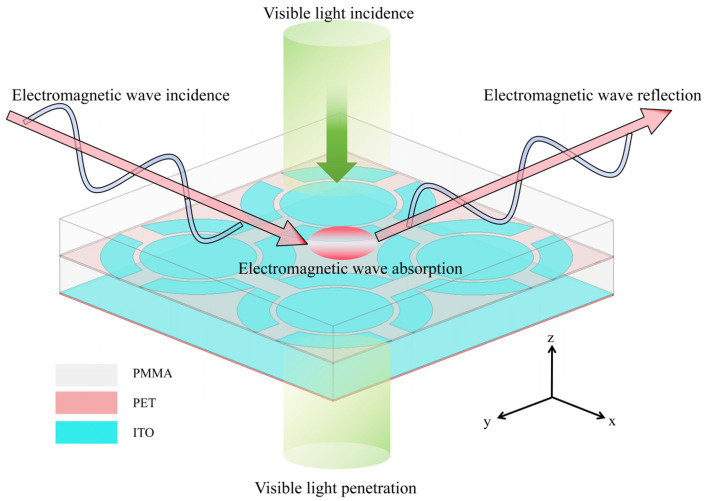
Schematic diagram of transparent absorber.

**Figure 2 micromachines-16-00653-f002:**
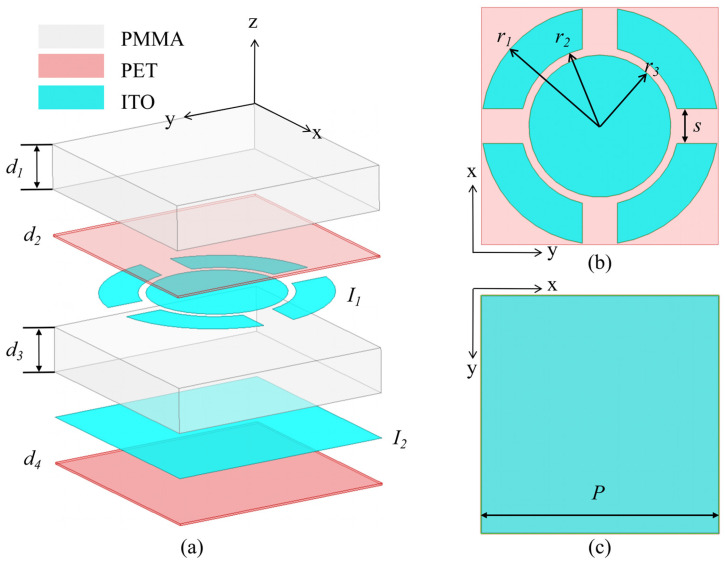
Absorber unit design. *P* = 15 mm, *r*_1_ = 7.5 mm, *r*_2_ = 5 mm, *r*_3_ = 4.5 mm, *s* = 2 mm, *d*_1_ = *d*_3_ = 3 mm, *d*_2_ = 0.125 mm, *d*_4_ = 0.175 mm. (**a**) Unit Exploded View; (**b**) Elevation view of the second layer of the absorber; (**c**) Top view of the fourth layer of the absorber.

**Figure 3 micromachines-16-00653-f003:**
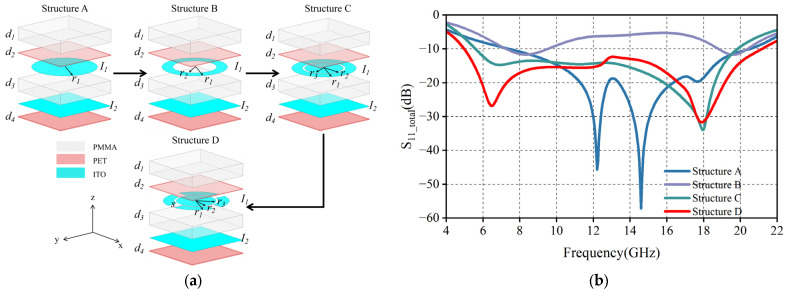
(**a**) MMA unit evolution process; (**b**) Reflection coefficients for different unit structures.

**Figure 4 micromachines-16-00653-f004:**
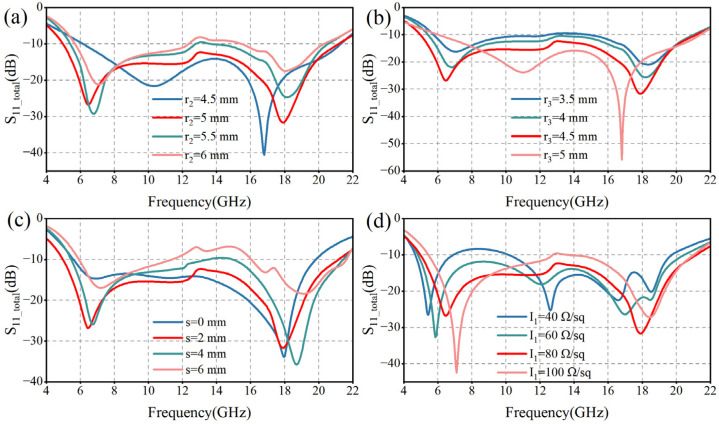
Reflection coefficients corresponding to different unit parameters of MMA. (**a**) Reflection coefficients for different *r*_2_ conditions; (**b**) Reflection coefficients for different *r*_3_ conditions; (**c**) Reflection coefficients for different *s* conditions; (**d**) Reflection coefficients for different *I*_1_ conditions.

**Figure 5 micromachines-16-00653-f005:**
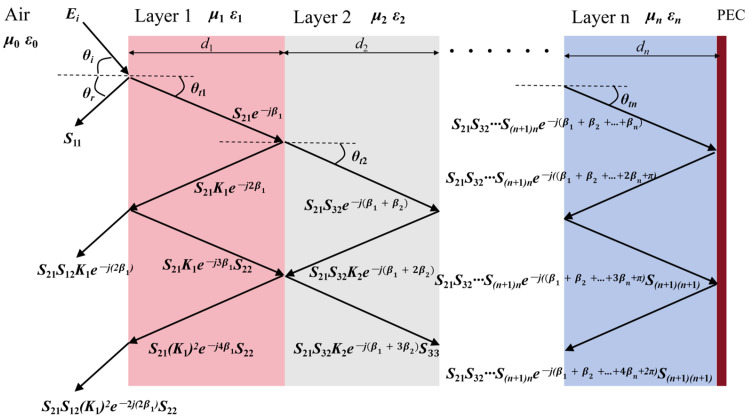
Electromagnetic wave propagation patterns based on multilayer dielectric interference theory.

**Figure 6 micromachines-16-00653-f006:**
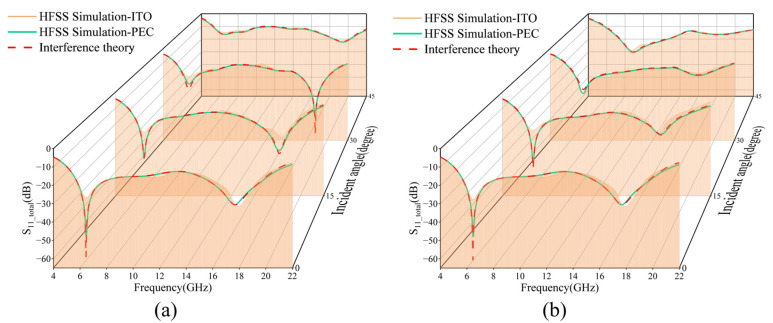
Comparison of HFSS Simulation and Interference theory results. (**a**) TM polarized wave 0–45° reflection coefficient; (**b**) TE polarized wave 0–45° reflection coefficient.

**Figure 7 micromachines-16-00653-f007:**
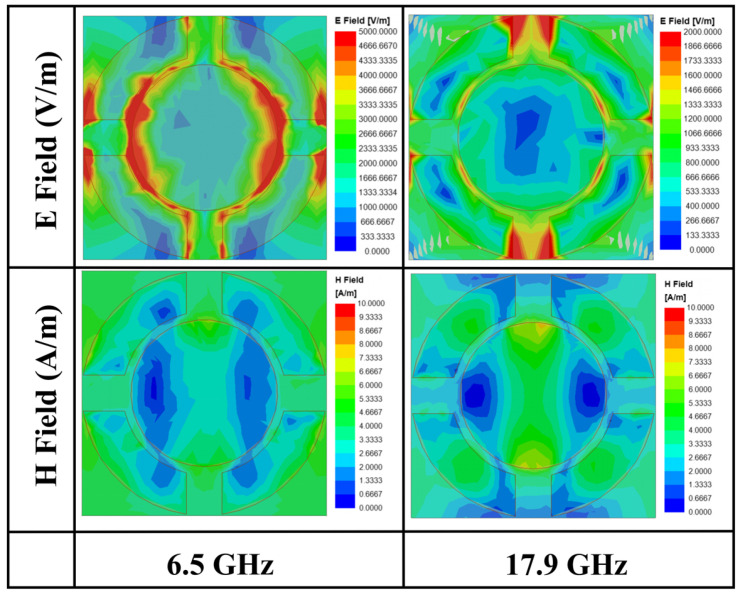
Simulated electric and magnetic field distributions at 6.5 GHz and 17.9 GHz.

**Figure 8 micromachines-16-00653-f008:**
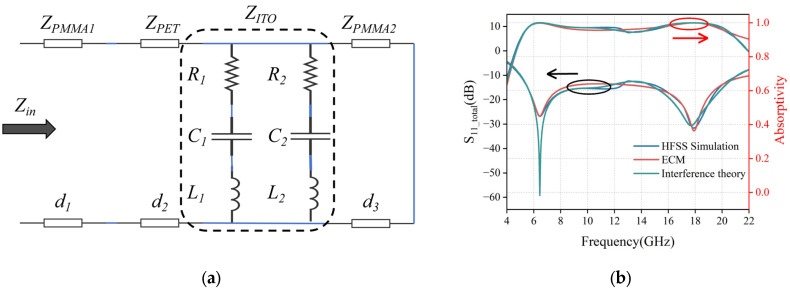
(**a**) ECM; (**b**) Comparison data plots of HFSS simulation results, ECM theoretical values, and interference theory values. (The direction of the arrow indicates the Y-axis of each data reference.)

**Figure 9 micromachines-16-00653-f009:**
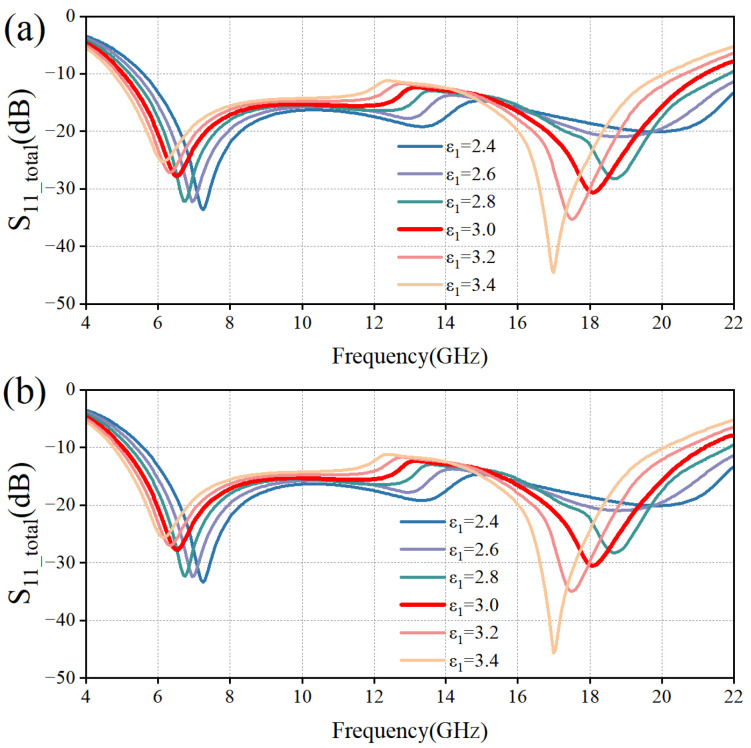
Effect of variation of *ε*_1_ on reflection coefficients of two polarizations. (**a**) TM Polarization; (**b**) TE Polarization.

**Figure 10 micromachines-16-00653-f010:**
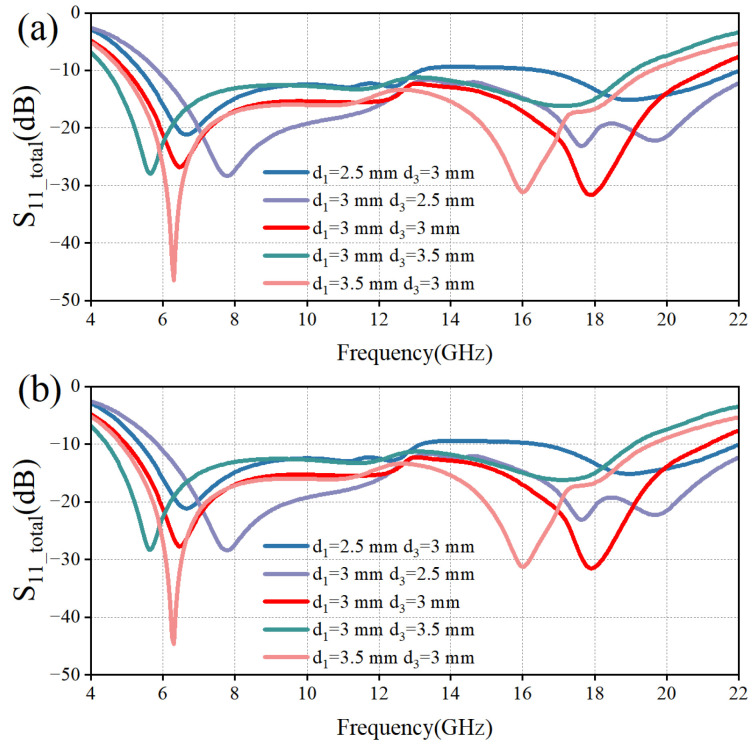
Effect of *d*_1_ and *d*_3_ variations on the reflection coefficients of the two polarizations. (**a**) TM Polarization; (**b**) TE Polarization.

**Figure 11 micromachines-16-00653-f011:**
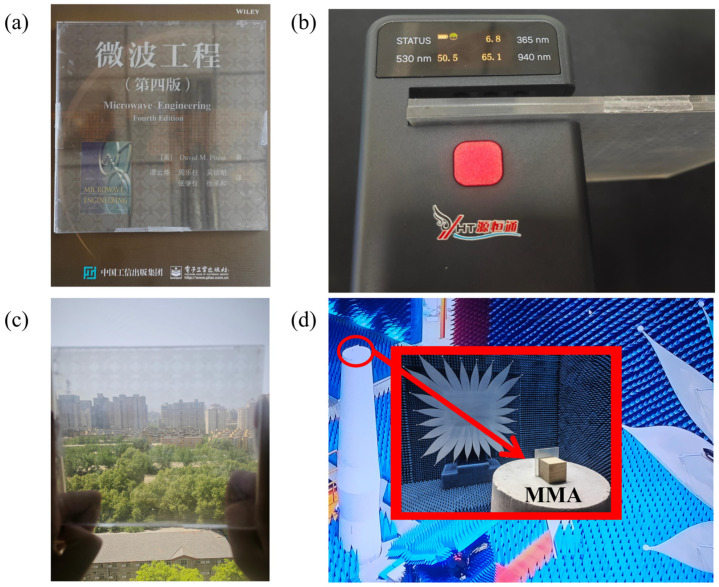
(**a**) Transparent MMA samples; (**b**) Schematic diagram of MMA single-point light transmittance test; (**c**) Schematic diagram of MMA light transmittance under sunlight; (**d**) MMA microwave darkroom test chart (The red box is a partial enlarged image).

**Figure 12 micromachines-16-00653-f012:**
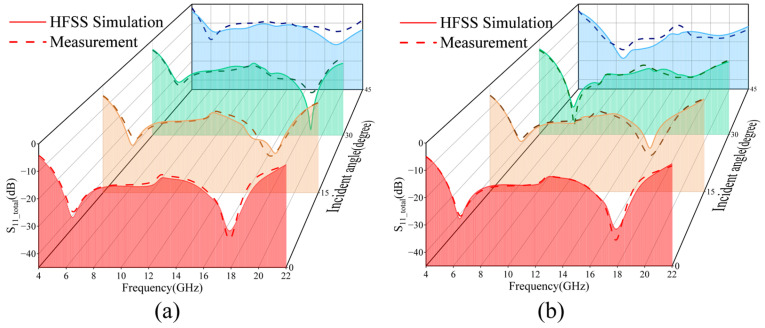
Simulation and experimental results of transparent wave absorber. (**a**) TM polarized wave 0–45° reflection coefficient; (**b**) TE polarized wave 0–45° reflection coefficient.

**Table 1 micromachines-16-00653-t001:** Expression of reflection and transmission coefficients.

Symbol	Explanation
*S* _(*n*−1)(*n*−1)_	The partial reflection coefficient of the wave from the (n − 1)-th layer when it is incident on the surface of the nth layer of the medium data
*S_nn_*	The partial reflection coefficient of the wave from the nth layer when it is incident on the surface of the (n − 1)-th layer of the medium
*S* _*n*(*n*−1)_	The partial transmission coefficient of a wave from the (n − 1)-th layer incident on the nth layer of the medium
*S* _(*n*−1)*n*_	The partial transmission coefficient of a wave from the nth layer incident on the (n − 1)-th layer of the medium

**Table 2 micromachines-16-00653-t002:** Comparison of Performance.

Document	Bandwidth (GHz)	FBW (%)	Structure Thickness	Angle Absorption		Transparency
				TM	TE	
[30]	2.03–6.98	109.9	0.08*λ_L_* (12 mm)	~45°	~45°	No
[31]	20.76–24.2	15.3	0.109*λ_L_* (1.575 mm)	~30°	~15°	No
[32]	6.54–18.66	96.2	0.117*λ_L_* (5.35 mm)	~50°	~50°	Yes
[33]	4.1–17.5	124	0.095*λ_L_* (7 mm)	~40°	~40°	Yes
[34]	8–18	76.9	0.12*λ_L_* (4.5 mm)	~30°	~30°	Yes
[35]	2–4.5	76.9	0.0847*λ_L_* (12.7 mm)	~30°	~40°	Yes
This work	5–21.15	123.5	0.105*λ_L_* (6.3 mm)	~45°	~45°	Yes

## Data Availability

All data generated or analyzed during this study are included in this manuscript. There are no additional data or datasets beyond what is presented in the manuscript.

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
