# Peer review of "Design and Analysis of Ultra-Thin Broadband Transparent Absorber Based on ITO Film"

_micromachines, 2025, doi:10.3390/mi16060653_

Round 1

Reviewer 1 Report

Comments and Suggestions for Authors

The title of the manuscript is inconsistent with the research content. The research content is very simple and basic, without any theoretical innovation or performance breakthrough. The English expression is poor, and the logical coherence of the research content is lacking, making it unsuitable for publication as a scientific research paper.

Reviewer 2 Report

Comments and Suggestions for Authors

In this work, the Authors present an optically transparent metamaterial absorber based on PMMA and patterned ITO layers that shows good absorption performance for a wide range of frequencies in the GHz band. Their design is backed by full-wave simulations, electromagnetic interference theory, and experimental measurements. In general, I think that the manuscript merits publication in Micromachines but I would like the Authors to revise the manuscript before publication according to the following comments.

1. I believe that section 2.1 is very brief. Specifically, I think that it would have been very interesting if the Authors elaborated a bit more on their design approach. Why do they use this patterning in ITO? What was their targeting and intuition? How did they ended up with the quoted design parameters? I strongly suggest to also include a discussion regarding the operating principle of their metamaterial absorber; why their design is so broadband, what effects do they capitalize on, etc. Maybe some new appropriate simulation results can be included as well.

2. On the contrary, I also believe that section 2.2 is too big. Their interference theory is not new but rather it consists of a textbook analysis of reflection/transmission in multilayered media. I did not find anything new or novel there so I suggest them either to shorten this presentation or highlight their new contribution, if there is any that I missed.

3. The results of Fig. 4 are very interesting. Their interference theory predicts larger reflection deeps that HFSS simulations. I guess this is due to the patterning that cannot be accurately captured in the former theory. However, I would like a) to see this better discussed in the manuscript and, importantly, b) I would like the authors to discuss is this interference theory can be applied to structures with different patterning. Do they expect to get equally good results? What are to constraints in general?

4. Which method do the Authors use to get the results of figures 8 and 9? Do they use interference theory? Firstly, this should be imprinted more clearly in the manuscript. And secondly, if this is indeed true, what are the benefits compared to simulations with HFSS. 

5. In figure 11, why do they compare the measurement results with HFSS and not their interference theory? Is there any other reason for not doing so? Can the latter be included as well for comparison?

6. Finally, the English in the manuscript are not that good and require revisions to meet the standards of the journal.

Comments on the Quality of English Language

As stated in my 6th point, I believe that the language of the manuscript merits revisions to be clearer and easily understandable.

Reviewer 3 Report

Comments and Suggestions for Authors

The authors proposed an ultrathin broadband transparent absorber based on ITO film, they chose PMMA instead of the traditional dielectric sheet and PET as the ITO film substrate. Simulation results indicated that the absorber achieves more than 90% absorption for positively incident electromagnetic waves in the broadband range of 5-21.15 GHz with a fractional bandwidth (FBW) of 123.5%. The work is interesting, and it can be recommended for publication after the following concerns have been addressed.

  1. Where does the primary absorption occur—in PMMA, PET, or ITO? Is the main cause of absorption the intrinsic loss of these materials themselves?
  2. The authors used the interference principle to explain the reasons for absorption. The manuscript listed six formulas. Do these formulas have a direct relationship with the design of this article?
  3. There are some metassurface-related works that are suggested to be discussed.

(1) Nano Letters 2020 20 (4), 2791-2798,

DOI: 10.1021/acs.nanolett.0c00471

(2) DOI: 10.1016/j.physleta.2024.129336

(3) Opto-Electron Adv 8, 240159 (2025). doi: 10.29026/oea.2025.240159
